# Increased IL-12p70 and IL-8 Produced by Monocytes in Response to *Streptococcus* spp. and *Actinomyces* spp. Causals of Endodontic Primary Infections

**DOI:** 10.3390/ijms242316853

**Published:** 2023-11-28

**Authors:** Raquel Sánchez-Gutiérrez, Janeth Araujo-Pérez, Diana Lorena Alvarado-Hernández, Ana María González-Amaro, Verónica Méndez-González, Bruno Rivas-Santiago, Roberto González-Amaro, Amaury Pozos-Guillén, Marlen Vitales-Noyola

**Affiliations:** 1Department of Immunology, Faculty of Medicine, Autonomous University of San Luis Potosí, San Luis Potosi 78290, San Luis Potosi, Mexico; r.sanchez@ttuhsc.edu (R.S.-G.); diana.alvarado@uaslp.mx (D.L.A.-H.); rgonzale@uaslp.mx (R.G.-A.); 2Department of Molecular and Translational Medicine, Paul L. Foster School of Medicine, Texas Tech University Health Sciences Center El Paso, El Paso, TX 79905, USA; 3Endodontics Postgraduate Program, Faculty of Dentistry, Autonomous University of San Luis Potosí, San Luis Potosi 78290, San Luis Potosi, Mexicoana.amaro@uaslp.mx (A.M.G.-A.); veronica.mendez@uaslp.mx (V.M.-G.); 4Biomedical Research Unit Zacatecas, Mexican Institute for Social Security-IMSS, Zacatecas 98000, Zacatecas, Mexico; bruno.rivas@imss.gob.mx; 5Basic Sciences Laboratory, Faculty of Dentistry, Autonomous University of San Luis Potosí, San Luis Potosi 78290, San Luis Potosi, Mexico; apozos@uaslp.mx; 6Department of Molecular Biomedicine, Center of Research and Advanced Studies of National Polytechnical Institute (CINVESTAV), Mexico City 07360, Mexico, Mexico

**Keywords:** monocytes, Interleukin-12p70, Interleukin-8, primary infection, *Streptococcus*, *Actinomyces*

## Abstract

We sought to evaluate the effect of endodontic-causative microorganisms of primary infections on mononuclear cells such as CD14^+^, CD4^+^, CD8^+^, CD19^+^ and Tregs Foxp3^+^. Facultative anaerobic microorganisms were isolated from radicular conducts and peripheral blood samples, which were taken from patients with primary infections. Cellular cultures were performed with peripheral blood mononuclear cells (PBMC) with and without *Actinomyces* spp. and *Streptococcus* spp. during 48, 72, and 96 h of contact in culture (concentration 5 × 10^5^ cells/well) in a round plate bound with 48 wells. Later, PBMC was collected for analysis by flow cytometry, with the monoclonal antibodies αCD14, αCD4, αCD8, αCD19 and αFoxp3, and acquired using an FACSCanto II cytometer. The supernatant of cellular cultures was analyzed for the quantification of inflammatory cytokines. Data analysis was performed in FlowJo v10.8.2 and FCAPArray software, and statistical analysis was performed using GraphPad v5.0. software. We observed an increase in the percentage of CD14^+^ cells in patients at different hours of cellular culture in the presence of both *Actinomyces* spp. and *Streptococcus* spp. microorganisms, compared to healthy controls. This study demonstrates the role played by the innate immune system in the pathogeny of endodontic primary infections, explaining the effects that generate the more common microorganisms in this oral pathology.

## 1. Introduction

In odontology clinical practice, pulp disease is the most frequent pathology, and the most frequent cause of dental consultations, due to the painful symptomatology [1]. Under healthy conditions, dental pulp and its periapical tissues are sterile; the dental pulp is a mesenchymal soft tissue that is highly vascularized and innervated, and has specialized cells (odontoblasts), which are in direct contact with the matrix of dentine. When the dental pulp suffers attacks, such as microbial, it can generate an immune response [2]. There are several access pathways for the bacterial invasion into pulp tissue: through dentinal tubules, principally by extensive and/or invasive carious, which is the most common pathway of infection, through an open cavity due to traumatisms and fissures, through the periodontal membrane due to the presence of microorganisms from gingival sulcus that can reach the pulp chamber through lateral canals or apical foramen, through blood via anachoresis, through contiguity as a consequence of the periapical infectious process of an adjacent tooth, etc. [3].

Endodontic infections are classified according to their anatomic localization into intraradiculars and extraradiculars. Intraradicular infections are divided into the ones where the microorganisms colonize the root canals, and those that are primary or secondary/persistent [4]. Out of the 700 microbial species that colonize the oral cavity, maybe 15 or 30 species have been detected inside the infected root canals [5].

Primary endodontic infections are caused by microorganisms that colonize the necrotic pulp tissue and represent the principal cause of apical periodontitis, which can be manifested as a chronic or acute disease. Their presence is the first step during the infection process of the dental organ, which affects the dental pulp and determines the further need for endodontic treatment. In primary infections, strict and/or facultative anaerobic bacteria predominate, and the most frequent species are Gram-negative (*Fusobacterium*, *Porphyromonas*, *Prevotella*, and *Veillonella*) and Gram-positive (*Actinomyces*, *Peptostreptococcus*, *Streptococcus*, *Propionibacterium*) [4].

The presence of microorganisms inside root canals causes dental pulp to induce an immune response [6], since any lesion, either physical, thermal, chemical, or of a microbial nature, triggers an inflammatory response. However, microorganisms are considered the principal etiological agent of pulp damage. When the pulp damage is a transitory issue, the inflammatory process is brief and subsides by itself, but when the irritant substances are present in excessive amounts or with persistent exposition, the specific and unspecific immune reactions cause the destruction of peri radicular tissues and the development of apical periodontitis. As carious lesions progress, the bacteria and their toxins enter the pulp space through dentinal tubules, and the odontoblasts detect bacteria toxins through Toll-like receptors (TLR) and contribute substantially to the pulp immune response. The dental pulp reacts to infection with an inflammatory tissue response [6].

The cells of the immune system are present in normal pulp and increase in number under pathologic conditions. The innate defense against microorganisms depends on several mechanisms, such as dentinal permeability, the presence of odontoblasts, cells of unspecific immunity (monocytes/macrophages, NK cells, and dendritic cells), and several soluble factors (cytokines). Of all these, the monocytes and later macrophages are critical in the defense of pulp, expressing pattern recognition receptors (PRR) for numerous bacterial molecules, acting as receptors for lipopolysaccharides, lipoteichoic acid, TLR4, and CD14, and allowing through their activation an appropriate immune response in the pulp [7].

Despite the role of microorganisms as direct causative factors of primary infections, the immune processes involved have not been fully elucidated. Infections in teeth quickly progress towards the pulp; however, it is not known exactly how the innate and adaptive responses develop inside the dental pulp.

The causative microorganisms of primary infections can stimulate the immune response; however, the roles played by different subpopulations of immune cells in the elimination of these microorganisms are unknown. In this study, we aimed to evaluate the effects of the microorganisms responsible for endodontic primary infections on the innate and adaptive immune cells to better understand the pathogenesis of the disease, ultimately contributing to the endodontic area.

## 2. Results

### 2.1. Levels of CD14^+^ Monocytes in the Endodontic Bacteria from Controls and Endodontic Primary Infection Patients

After the stimulation of PBMC in culture with *Actinomyces* spp. and *Streptococcus* spp. over several durations of contact, we observed increases in these cells at 48, 72, and 96 h after the exposition in both the control and patients compared to levels at zero hours—Figure 1B,C (controls—7.27, 3.82–10.35%, 21.20, 17.15–34.0%, 27.2, 22.8–35.0%, 38.7, 30.3–43.9%, median and interquartile range, 0, 48, 72, and 96 h of incubation time, respectively, *p* < 0.05; patients—6.26, 3.69–11.17%, 23.5, 18.37–28.55%, 37.0, 26.7–40.4%, 30.35, 23.75–37.4%, median and interquartile range, 0, 48, 72, and 96 h of incubation time, respectively). However, when we compared data without considering the time of incubation (48, 72, and 96 h) between the two tested microorganisms, no significant differences between the groups analyzed were found (Figure 1D, *p* = 0.64). In addition, we evaluated the cells present at low numbers for CD14, and we observed an increase in the percent of CD14^lo^ cells in PBMC from patients stimulated with *Actinomyces* spp. at 48, 72 and 96 h in comparison to healthy controls (18.95, 15.95–28.25%, 20.5, 17.5–29.6%, 23.3, 14.5–24.0%, median and interquartile range, 48, 72, and 96 h of incubation time, respectively, data from patients); *p* < 0.05, Figure 1E. Similar results were observed in PBMC from patients stimulated with *Streptococcus* spp. (13.0, 12.01–26.0%, 18.55, 15.93–20.03%, median and interquartile range, 72, and 96 h of incubation time, respectively, data from patients); *p* < 0.05, Figure 1F. The flow cytometry staining strategy is shown at the beginning of the figures, with a representative example in Figure 1A.

### 2.2. Levels of CD4^+^ and CD8^+^ Lymphocytes in the Presence of Endodontic Bacteria from Controls and Endodontic Primary Infections Patients

After the stimulation of PBMC in culture with *Actinomyces* spp. over several incubation durations, we observed an increase in CD4^+^ lymphocytes at 48 and 72 h after exposure in comparison to controls at time zero—Figure 2B (31.1, 27.7–36.15%, 39.65, 37.45–52.1%, 43.7, 38.8–47.9%, median and interquartile range, 0, 48 and 72 h of incubation, respectively, *p* < 0.0001). No significant differences at 96 h post-incubation were observed (42.7, 32.4–46.3%, median and interquartile range). In the patient samples, we did not observe significant differences in the percentages of CD4^+^ lymphocytes in the cultures of several different conditions (28.1, 22.5–36.88%, 39.1, 31.85–44.03%, 37.3, 34.28–39.1%, 34.0, 29.88–42.1%, median and interquartile range, 0, 48, 72, and 96 h of incubation time, respectively, *p* > 0.05). When PBMC were stimulated with *Streptococcus* spp., we observed an increase in the percent of CD4^+^ lymphocytes at 48, 72, and 96 h of incubation in controls, in comparison to zero hours (31.1, 27.7–36.15%, 35.2, 28.8–45.1%, 48.1, 46.1–49.98%, 45.1, 42.63–48.68%, median and interquartile range, 0, 48, 72, and 96 h of incubation, respectively, *p* < 0.0001). This contrasts with the percentages shown by patients, where we only observed an increase at 72 h in comparison to zero hours (29.95, 23.28–40.38%, 39.1, 33.8–49.6%, median and interquartile range, 0 and 72 h, respectively, *p* < 0.0; Figure 2B,C). In contrast, regarding the CD8^+^ lymphocytes stimulated with *Actinomyces* spp., we observed them to be increased only at 96 h of incubation in comparison to zero hours in controls (7.17, 6.19–9.29%, 42.7, 32.7–46.0%, median and interquartile range, 0 and 96 h, respectively, *p* < 0.0001), while in patients, no significant differences were observed—Figure 2D. Finally, for the CD8^+^ lymphocytes that were stimulated with *Streptococcus* spp., no significant differences between the two groups under several culturing conditions were observed (*p* > 0.05)—Figure 2E. The flow cytometry staining strategy is shown at the beginning of the figures, with a representative example shown in Figure 2A.

### 2.3. Levels of CD19^+^ B Lymphocytes in the Presence of Endodontic Bacteria from Controls and Endodontic Primary Infection Patients

After stimulation of PBMC in culture with *Actinomyces* spp. during several contact times, as described in the Section 4, we did not observe a difference in the percent of CD19^+^ B lymphocytes under the different tested culture conditions, Figure 3B, (5.62, 4.87–7.82%, 8.66, 6.12–10.02%, 8.85, 6.52–10.9%, 7.33, 5.75–11.03%, median and interquartile range, 0, 48, 72 and 96 h of incubation, respectively, *p* > 0.05, data from controls), (10.7, 7.6–13.8%, 9.26, 7.52–9.89%, 12, 9.57–13.9%, 9.24, 6.64–12.25% median and interquartile range, 0, 48, 72, and 96 h of incubation, respectively, *p* > 0.05, data from patients). When PBMC were stimulated with *Streptococcus* spp., we observed an increase in the percent of CD19^+^ B lymphocytes only at 72 h of incubation in controls, in comparison to zero hours (5.62, 4.87–7.82%, 8.28, 6.05–9.1%, 8.8, 7.08–14.03%, 6.23, 6.09–9.26%, median and interquartile range, 0, 48, 72, and 96 h of incubation time, respectively, *p* < 0.0001), Figure 3B, in contrast with the percent from patients, where no differences at any of the tested conditions were observed. (10.72, 7.69–13.8%, 7.47, 6.46–12.48%, 12.7, 9.27–15.4%, 10.06, 7.8–11.73% median and interquartile range, 0 and 72 h, respectively, *p* > 0.05) Figure 3C. The flow cytometry staining strategy is shown at the beginning of the figures, with a representative example, Figure 3A.

### 2.4. Levels of Foxp3^+^ T Regulatory Cells in the Presence of Endodontic Bacteria from Controls and Endodontic Primary Infection Patients

After the stimulation of PBMC in a culture with *Actinomyces* spp. over several contact durations, as described in the Section 4, no significant differences in the percent of Foxp3^+^ Tregs lymphocytes under the different culturing conditions were observed—Figure 4B (controls—1.69, 1.36–2.08%, 2.09, 1.21–3.09%, 2.25, 1.92–3.58%, 2.25, 1.71–3.27%, median and interquartile range, 0, 48, 72 and 96 h of incubation time, respectively, *p* > 0.05; patients—1.45, 1.13–2.07%, 1.93, 1.4–2.47%, 2.09, 1.42–2.53%, 1.93, 1.58–2.83% median and interquartile range, 0, 48, 72, and 96 h of incubation time, respectively, *p* > 0.05). When PBMC were stimulated with *Streptococcus* spp., we observed an increase in the percent of Foxp3^+^ Tregs lymphocytes only at 72 h of incubation in controls, when compared to zero hours of incubation (1.69, 1.36–2.08%, 2.03, 1.71–2.89%, 2.54, 1.59–3.63%, 2.26, 1.53–2.66%, median and interquartile range, 0, 48, 72, and 96 h of incubation, respectively, *p* < 0.0002), in contrast with the percent of FoxP3 in patients’ cellular cultures, wherein we did not observe significant differences under any of the tested conditions (1.45, 1.13–2.07%, 1.8, 1.42–2.54%, 3.56, 1.97–3.82%, 2.68, 1.53–3.23% median and interquartile range, 0 and 72 h, respectively, *p* > 0.05)—Figure 4C. The flow cytometry staining strategy is shown at the beginning of the figures, with a representative example shown in Figure 4A.

### 2.5. Quantification of Proinflammatory Cytokines in Supernatant of Cellular Cultures

Finally, we analyzed the levels of the pro-inflammatory cytokines IL-1β, TNF-a, IL-6, IL-8, IL-10, and IL-12p70 in the supernatants of the cell cultures, as described in the Material and Methods. We observed a significant increase in the release of IL-12p70 in the supernatants of cellular cultures from the patients’ samples when stimulated with *Streptococcus* spp. at 48, 72, and 96 h of culture (*p* < 0.05, compared to control condition cultures)—Figure 5A. In addition, we observed an increase in the release of IL-8 when the PBMC was stimulated with both microorganisms, *Actinomyces* spp. and *Streptococcus* spp. (*p* < 0.0015 and *p* < 0.0056, respectively, compared to control condition cultures)—Figure 5B,C. For the other cytokines (IL-1β, TNF-a, IL-6, and IL-10), no significant differences were observed.

## 3. Discussion

The inflammation of periarticular tissues can be caused by endodontic primary or secondary/persistent infections. These infections are caused by the presence and persistence of microorganisms inside root canals. The severity of infection depends on the virulence of microorganisms, the affected anatomic area, and the immune response of the host [8]. Regarding primary infections, polymicrobial colonization can be related to early phases of invasion in the dental pulp, culminating in inflammation and posterior necrosis. The resolution of these primary infections depends on the response of immune cells, since an infected pulp responds to microorganisms in the apical sense, stimulating the inflammatory and immunological response, the participation of monocytes and lymphocytes in which has been described. Infections in dental pulp are mainly regulated by monocytes, which are mononuclear circulating phagocytes originating in the bone marrow that give rise to macrophages in peripheral tissue; however, as the infection progresses, the participation of cells in the immune-specific response is essential, such as T and B lymphocytes [7,9].

Monocytes/macrophages are cells of the primary immune response that act against a great number of harmful stimuli and can migrate to tissues, ultimately quickly removing the stimulus and generating more specific immune responses [10]. Previous studies reported an increase in these cells during the inflammatory process of the dental pulp [11,12,13], and these findings agree with our results, which may be due to the presence of monocytes with proliferative capacities due to the release of M-CSF by endodontic microorganisms, or simply to the non-survival of the rest of the PBMC in the culture.

In this study, *Streptococcus* spp. and *Actinomyces* spp. significantly activated monocytes, both in patients and in healthy subjects, when the time of exposition between PBMC/bacteria was short (48 and 72 h). This is probably due to the exhaustive activation of monocytes being favored by CD4^+^ T lymphocytes, and the production of inflammatory cytokines such as IFN-γ; however, this mechanism is only theoretical, and is of interest for future research. In this respect, it has been described that *Streptococcus mutans* is a cariogenic bacterium that induces polarization to Th1 lymphocytes (great producers of IFN-γ) [14], which could explain the increased levels of monocytes and CD4^+^ T lymphocytes in PBMC from patients stimulated with *Streptococcus* spp. However, no correlation between the rates of activation of monocytes by *Actinomyces* spp. was observed, which suggests the activation of an independent pathway in Th1 cells and IFN-γ; however, the design of new studies elucidating the cellular and molecular mechanisms involved is necessary. In addition, we observed increases in the percentages of proliferative monocytes (PMs) CD14^lo^ in both groups (controls and patients). The heterogeneity of peripheral blood monocytes has led to the notion that different monocyte subpopulations may have special or restricted functions. For example, in humans, there is a minor subpopulation of monocytes—CD14^lo^ CD16^+^—that has been implicated in several inflammatory pathologies. Human monocytes are commonly considered to be non-proliferating [15]; however, a subpopulation of human monocytes that is capable of proliferating in vitro (for example, in response to M-CSF) has been described. This population has been referred to as proliferative monocytes (PMs), which were recently shown to have the phenotype CD14^+^ CD16^−^ CD64^+^ CD33^+^ CD13^lo^ c-Fms^+^ [15]. It was previously suggested that PMs might be able to migrate into inflamed tissues, and possibly undergo local proliferation [15].

We observed increased levels of CD4^+^ lymphocytes in patients after stimulation with *Streptococcus* spp., which might suggest that the patients in this study experienced chronic primary infections, or that the employed microorganisms did not generate the same response as other microorganisms, such as *S. mutans*. There is evidence that indicates that *Actinomyces viscosus* generates mixed responses, both inflammatory and anti-inflammatory, with a greater tendency towards the anti-inflammatory phenotype, especially for IL-2-producing T CD4^+^ lymphocytes (Th2) [16], which suggests that a regulatory immune response is dependent on a particular microorganism. However, more information in this respect is yet to be elucidated. We observed increased levels of CD8^+^ T lymphocytes in PBMC from patients after 72 h of incubation in the presence of *Streptococcus* spp., in comparison with controls without bacteria, and it has been described that *S. mutans* can induce the production of IFN-γ by CD8^+^ T lymphocytes [17]. A pressing issue to address is whether the high percentages of CD8^+^ lymphocytes in the first stages can promote the polarization of IFNγ-producing CD4^+^ lymphocytes (Th1), which could explain the increase in these cells in the patients. In contrast, in the PBMC of healthy subjects stimulated with *Actinomyces* spp., we observed a remarkable increase in CD8^+^ and CD4^+^ T lymphocytes, accompanied by a trend toward the increase in Foxp3^+^ cells in healthy subjects, which could suggest the possible regulation between these effector cells.

The participation of B lymphocytes in multiple pulp bacterial infections has been previously described; these cells recognize bacterial antigens and act as antigen-presenting cells, being part of the first adaptive immune response. Several studies have identified active B lymphocytes in endodontic lesions [18,19]. In addition, in this study, we observed a high percentage in a basal manner in patients with primary infections (time zero of incubation), which suggests the participation of these cells with active or memory phenotypes in circulation, probably originating from the site of the lesion, since these cells tend to have a long life and persist in the organism [20]. However, in this study, we could not observe an increase in this cell population after the stimulation with *Actinomyces* spp. or *Streptococcus* spp., which could have been due to a short stimulation time in the presence of the pathogens, since these cells are part of the humoral adaptive immunity and thus require more time to act; the specificity of its functions; a specific pathway not being describable under our study conditions; or the death of these cells in culture. These results are contrary to what we expected, since the B lymphocytes are the first cells to show a more effective adaptive response compared to innate response.

The regulatory cells are characterized by the expression of the transcription factor Foxp3, thus constituting a unique subpopulation that inhibits the immune response and thus checks the exacerbation of inflammation during infection. Several studies have found an increase in Foxp3^+^ T lymphocytes in endodontic lesions [21,22]; however, in this study, we did not observe significant differences between the groups stimulated with *Actinomyces* spp. or *Streptococcus* spp. at the different tested times (0, 48, 72, 96 h). The Foxp3^+^ lymphocytes are part of the adaptive immune response, which is slower compared to the innate response, taking up to several weeks to develop. Even though we found an increasing trend from 0 to 96 h, the analysis duration selected in the present study might not be sufficient to detect this cellular phenotype, or these cells could have died in the culture. In this regard, Alswhaimi et al. [21] suggested that the action of regulatory T cells is time-dependent, and can be found from 7 to 14 days after infection in mice with periapical lesions [21]. It would be of interest for future studies to determine the regulatory response over a longer period.

On the other hand, the close relationship between regulatory T cells and Th17 lymphocytes is well-known [23]. We observed increased levels of CD4^+^ lymphocytes in cells from patients stimulated with *Streptococcus* spp.; therefore, a further study should be undertaken, exploring whether the existing cytokines preferentially favor the differentiation of Th17 cells over regulatory T cells.

Cytokines are key modulators of the homeostasis and inflammatory processes involved in the initial response against pathogens. We found increased levels of IL-12p70 and IL-8 in supernatants of PBMC stimulated with *Streptococcus* spp. IL-12p70 is a cytokine that is involved in the activation of T cells and is produced by monocytes/macrophages and dendritic cells [24]; this increase might explain our increased levels of T cells. It has been described that some strains of streptococcus-mediated IL-12p70 induced the activation of Toll-Like Receptor 3 in dendritic cells [25,26]. In contrast, this cytokine is known as a potent inductor of IFN-γ secretion and Th1 differentiation, and as an important suppressor of Th2 cytokines, such as IL-4 [27,28,29]. This immunological interaction could support the inflammation and destruction of periapical tissues. In this regard, significant correlations were observed between levels of IL-1α and Th1-derived pro-inflammatory mediators IL-2, IL-12, TNF-α, and IFN-γ [30]. These findings suggest that IL-12 may play a role in promoting the release of inflammatory molecules while inhibiting Th2 cytokines. This could potentially contribute to the development and worsening of infection-induced periapical lesions. In contrast, the amounts of IL-12 in the gingival crevicular fluid of chronic periodontitis patients were similar to those in healthy controls, and increased after periodontal therapy [31,32]. Studies indicate that IL-12p70 and IFN-γ can suppress osteoclastogenesis and bone resorption in vitro and in vivo [33]. These data suggest the role of IL-12 in tissue remodeling following inflammatory processes. We observed a sustained increase in this cytokine over time, which suggests its important role in periodontal pathogenesis. However, the dichotomous role of IL-12p70 during primary infections and progression, and the role of isolated species, would be interesting points for further elucidation.

IL-8 is rapidly synthesized at the local sites of inflammation to recruit, activate, and retain inflammatory cells, including monocytes and macrophages [34]. Our findings of increased IL-8 might be associated with the different bacterial strains present in the infected root canal, including *Streptococcus* spp. and *Actinomyces* spp. [35]. In this regard, an increase in cytokines as part of the innate immune response, as well as an increase in the frequency of monocytes, was found, suggesting that these cells could act as key regulators in primary infections. In addition, it has been described that IL-8 can be detected in monocytes between 2 and 6 h post-infection, and it is dose-dependent [36,37]. In contrast, IL-8 is involved in the recruitment and activation of osteoclasts, which are responsible for bone resorption in apical periodontitis, a possible complication of primary infections. Therefore, IL-8 is an important biomarker for the identification and management of infectious diseases.

This study was limited in the following areas, which thus represent options for further study: the correlational analysis between the clinical parameters and the numbers and/or functions of several immune cells; the evaluation of a large number of patients, or patients with other endodontic pathologies, to determine the soluble components related to the inflammation process; the immune phenotyping of more complex cellular populations with a large number of parameters, as well as several subtypes of regulatory T cells, effector cells such as Th1, Th2 and Th17, and other subpopulations of the innate immune system, such as dendritic cells. It would also be interesting to analyze the absolute numbers of these cellular subpopulations with data related to blood biometry.

## 4. Materials and Methods

### 4.1. Patients

Twenty patients with primary endodontic infections attending the Endodontic service were included in this study. All patients were subjected to oral examinations and gave their clinical history; patients with previous endodontics treatment, chronic degenerative and/or autoimmune diseases, or the presence of recent gastrointestinal, dermic, urinary, throat, etc., infection (at least 3 months) were excluded. In addition, twenty-five healthy subjects without endodontic infections were included as controls, with ages and genders similar to those of patients. This project was approved by the Ethical and Research Committee of Dentistry Faculty, of the Autonomous University of San Luis Potosi, with the code CEI-FE-034-019. All patients and controls signed an informed consent form.

### 4.2. Samples

Peripheral blood (6 mL) was taken from both patients and controls and placed in an ethylene–diamine–tetra acetic acid (EDTA) tube for the isolation of mononuclear cells through gradient density, employing Ficoll–Paque (GE Healthcare, Uppsala, Sweden). Cellular viability was evaluated by microscopy through trypan blue staining. Samples of root canals were taken for the isolation of endodontic microorganisms, introducing a file or #20 paper point into root canals with a larger caliber for 1 min. Previously, local anesthesia was applied to the region to be treated with mepivacaine 2% and epinephrine 1:100,000 (Scandonest, Septodont Saint-Maur-des-Fossés, France). Absolute isolation with a rubber dam, clamp, and Young’s arch was employed for endodontic access to the pulp chamber, which was later covered with a sterile cotton swab. Operating field disinfection was performed (hydrogen peroxide 30% for 1 min, sodium hypochlorite 5.25% for 1 min, and sodium thiosulphate 10% for 1 min); to check the adequate sterility of the operative field and avoid cross-contamination, a culture in trypticase soy agar (BD BBL, Sparks, MD, USA) at 37 °C for 24–48 h (FE-1320, Felisa, Jalisco, MX, México) was performed. The cotton swab was removed from the pulp chamber, and files or paper points were introduced into this compartment to be later placed in a transport medium (pre-reduced thioglycolate broth) (BD BBL, Edo de México, MX, México) for incubation in an anaerobiosis chamber (COY Laboratory Products, Inc., Grass Lake, MI, USA).

### 4.3. Isolation and Microorganisms’ Identification

The samples with files or paper tips of root canals were incubated under anaerobic conditions with 85% nitrogen, 10% hydrogen, and 5% carbon dioxide in an anaerobic chamber at 37 ± 2 °C for 48–72 h or until visible microbial growth. Microbial density was measured in a McFarland densitometer (Densimat; bioMérieux, Florence, Italy). Later, the samples from the tubes were spread on blood anaerobic agar (BD BBL, Edo de México, MX, México) under anaerobic conditions for 48–72 h or until visible microbial growth. The description of the macro- and microscopic features of microbial cultures was undertaken with stereoscopic microscopy (Leica EZ4D; Singapore) and Gram staining (HYCEL, Jalisco, MX, México), while also evaluating spore presence. The microbial identification was performed through biochemical tests employing a semi-automatized API system (bioMérieux, SA, Marcy l’Etoile, France) with the use of APIWeb v2.0 software (bioMérieux, SA, Marcy l’Etoile, France) in accordance with manufacturing instructions, using API 20 Strep for Streptococci, the related API 20 A for anaerobic coccus and bacillus, and API 20C AUX for yeasts.

### 4.4. Phagocytosis Assays

Phagocytosis assays were performed on the PBMC from five healthy controls and patients, analyzing the internalization of carboxy-fluorescein (CFSE) (ThermoFisher Scientific, Waltham, MA, USA). Bacteria were labeled with 5.0 μM of CFSE for 20 min at 4 °C; several different quantities of labeled *Streptococcus* spp. were tested (multiplicity of infection, MOI = 10, MOI = 100 and MOI = 1000) and different concentrations of PBMC (1.0 × 10^6^, 5.0 × 10^5^, 2.5 × 10^5^, 1.2 × 10^5^, and 6.5 × 10^4^) were mixed and incubated for 2.0 h at 37 °C. Negative controls of the assay were performed without bacteria. Then, the cells were washed, centrifuged at 1500 rpm for 5 min, and analyzed in an FACSCanto II flow cytometer (BD, San Diego, CA, USA), using the FlowJo software v10.0 (BD, San Diego, CA, USA). Cellular viability was evaluated by trypan blue staining, before and after phagocytosis assays.

### 4.5. Cellular Cultures

The peripheral blood mononuclear cells (PBMC) isolated from patients and controls were collocated in a 48-well plate at a concentration of 5 × 10^5^ cells/well in RPMI 1640 medium (Gibco, Grand Island, NY, USA) with MOI (multiplicity of infection) = 10, according to the previous assay described in the last point. Several culturing conditions were employed. Firstly, PBMC with *Actinomyces* spp. or *Streptococcus* spp. were collocated at 37 °C with 5% of CO_2_ in a plate with the bacterial strains, with a time of contact of two hours, and these were later discarded after several washes with RPMI 1640 medium. Non-phagocyted extracellular bacteria were eliminated by incubation with a medium containing ciprofloxacin (8 μg/mL) for 1 h. The validity of this process was confirmed by the absence of bacterial growth when inoculated on brain heart infusion (BHI) agar [38]. The culture times that were employed for each assay were 48, 72, and 96 h. Negative controls were constructed in all culture conditions, comprising cells without any type of stimulus and/or bacteria. Subsequently, at a determined time (48, 72, and 96 h), the cells were collected into cytometry tubes for analysis by flow cytometry, while the supernatant was collected in cryogenic tubes and stored at −80 °C for the posterior quantification of human inflammatory cytokines using a cytometric bead array (CBA) I Kit (BD Biosciences, Franklin Lakes, NJ, USA).

### 4.6. Cellular Staining and Flow Cytometry Analysis

The cells of cultures developed under several conditions were collocated in cytometry tubes, and were stained with the monoclonal antibodies α-CD14/FITC (BD Horizon, San Jose, CA, USA), α-CD4/APCCy7 (BD Biosciences, San Jose, CA, USA), α-CD8/APC (BioLegend Inc., San Diego CA, USA) and α-CD19/PE (eBioscience Inc., San Diego, CA, USA) for 30 min at 4 °C. For intracellular staining with α- Foxp3/PE Cy7 (BD Biosciences), the cells were fixed and permeabilized with Fix/Perm kit, according to the manufacturing instructions (eBioscience). Negative controls of all the fluorochromes employed were designed according to the fluorescence minus-one (FMO) strategy, and cells were analyzed in a FACSCanto flow cytometer (Becton–Dickinson, San Diego, CA, USA) acquiring at least 100,000 events/samples, and data analysis was performed using FlowJo software v10.0 (Tree Star Inc., Ashland, OR, USA).

### 4.7. Quantification of Inflammatory Cytokines

In the supernatants of cellular cultures, the quantification of human inflammatory cytokines was performed using the Cytometric Beads Array (BD Biosciences) kit according to the manufacturer’s instructions, which included the following cytokines: IL-8, IL-1β, IL-6, IL-10, TNF and IL-12p70. The samples were acquired via FACSAccuri flow cytometry (BD Biosciences) and the analysis was performed using the FCAPArray v3.0 software (BD Biosciences). Negative controls of supernatants were developed under all culture conditions, comprising cells without any stimulus and/or bacteria.

### 4.8. Statistical Analysis

Statistical analysis was performed using GraphPad Prism v5.0 software (GraphPad Software, San Diego, CA, USA). Data with normal distributions were represented as the arithmetic mean and SD, and data with a non-Gaussian distribution were represented as the median and interquartile range (Q1–Q3). Comparisons of the two groups were undertaken with the t-Student or the Mann–Whitney U tests. Comparisons among three groups were undertaken via the Kruskal–Wallis non-parametric test, or via ANOVA, with the post hoc analysis of Dunn’s post-test or Tuckey’s test, respectively. *p* values < 0.05 were considered significant.

## 5. Conclusions

Our results suggest a relation between immunocompromised organisms in patients with primary endodontic infections, directed mainly by monocytes and their secreted cytokines such as IL-12p70 and IL-8, and the presence of some of the causal microorganisms of this pathology (*Actinomyces* spp. and *Streptococcus* spp.) in the localized site of the infection. In addition, we have described for the first time in this pathology the participation of monocytes with a CD14^low^ phenotype. However, there are still many points to be elucidated, amongst which are the nature of the pathology during the establishment of the primary infection, and the cellular and molecular mechanisms involved. In addition, studies focused on the immune responses of secondary/persistent infections could also be of interest, since we partially understand the role that the immune response plays in primary endodontic infections. In addition, how the state of chronicity is established, as well as how the severity of infections, the development of the disease, and the persistence of causal microorganisms are influenced by the immunological response, also remain to be elucidated.

## Figures and Tables

**Figure 1 ijms-24-16853-f001:**
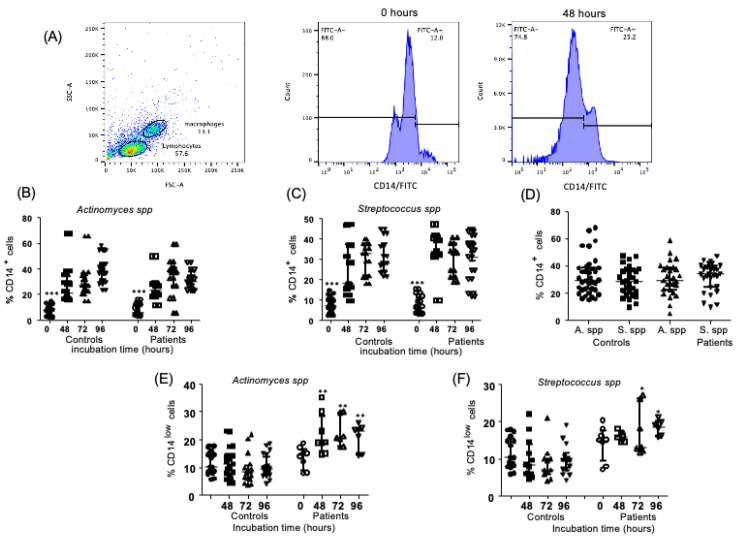
Percent of CD14^+^ cells after stimulation with endodontic microorganisms. The culturing of PBMC and endodontic bacteria was performed at different times, and staining was performed with flow cytometry analysis, as indicated in the Section 4. (**A**) Representative dot plots of a healthy subject (control) at 0 and 48 h of stimulation with endodontic microorganisms, showing the histograms corresponding to CD14^+^ cells stained with isothiocyanate of fluorescein (FITC). (**B**) Percentages of CD14^+^ cells in controls and patients with primary infections, stimulated with *Actinomyces* spp. at 0, 48, 72, and 96 h of culture. (**C**) Percentage of CD14^+^ cells in controls and patients with primary infections, stimulated with *Streptococcus* spp. at 0, 48, 72, and 96 h of culture. (**D**) Comparison of percentage of CD14^+^ cells in controls and patients with primary infections stimulated with *Actinomyces* spp. and *Streptococcus* spp. at all the tested incubation times. (**E**) Percent of CD14^lo^ cells in controls and patients with primary infections, stimulated with *Actinomyces* spp. at 0, 48, 72, and 96 h of culture. (**F**) Percent of CD14^lo^ cells in controls and patients with primary infections, stimulated with *Streptococcus* spp. at 0, 48, 72, and 96 h of culture. (**B**–**F**) Data correspond to the median and interquartile range, *n* = 25, * *p* < 0.05. ** *p* < 0.01. *** *p* < 0.001.

**Figure 2 ijms-24-16853-f002:**
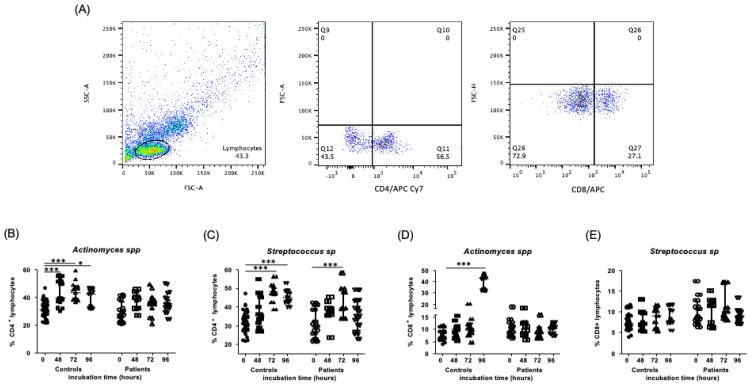
Percent of CD4^+^ and CD8^+^ lymphocytes after stimulation with endodontic microorganisms. The PBMC and endodontic bacteria cultures were performed at several incubation times and stained by flow cytometry analysis, as indicated in the Section 4. (**A**) Representative dot plots of a healthy subject (control) stimulated with endodontic microorganisms, showing CD4^+^ lymphocytes stained with allophycocyanin Cy7 (APC-Cy7) and CD8^+^ lymphocytes stained with allophycocyanin (APC). (**B**) Percent of CD4^+^ cells in controls and patients from primary infections, stimulated with *Actinomyces* spp. at 0-, 48-, 72-, and 96-h incubation. (**C**) Percent of CD4^+^ cells in controls and patients from primary infections, stimulated with *Streptococcus* spp. at 0, 48, 72, and 96 h of culture. (**D**) Percent of CD8^+^ cells in controls and patients from primary infections, stimulated with *Actinomyces* spp. at 0, 48, 72, and 96 h of culture. (**E**) Percent of CD8^+^ cells in controls and patients from primary infections, stimulated with *Streptococcus* spp. at 0, 48, 72, and 96 h of culture. Data correspond to the median and interquartile range, *n* = 25, * *p* < 0.05. *** *p* < 0.001.

**Figure 3 ijms-24-16853-f003:**
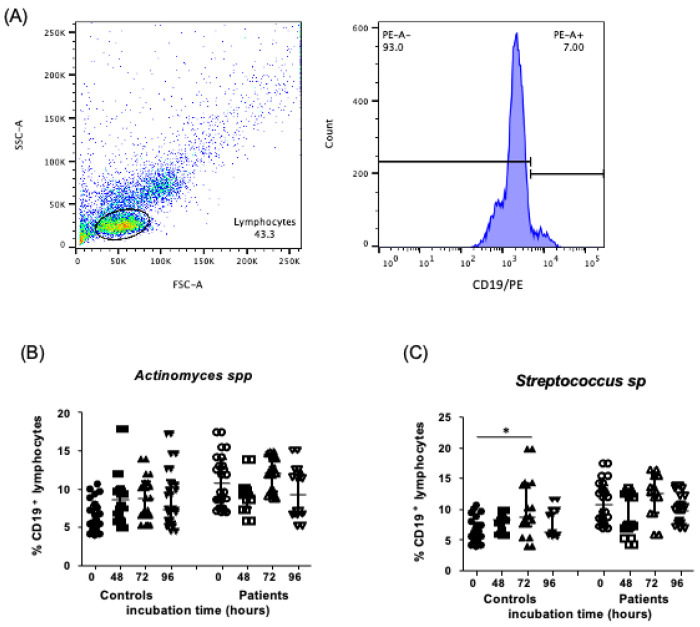
CD19^+^ B lymphocytes after stimulation with endodontic microorganisms. The PBMC and endodontic bacteria cultures were performed over several incubation times and stained as part of the flow cytometry analysis, as indicated in the Section 4. (**A**) Representative dot plots of a healthy subject (control) stimulated with endodontic microorganisms, where CD19^+^ lymphocytes stained with phycoerythrin (PE) are shown. (**B**) Percent of CD19^+^ cells in controls and patients with primary infections, stimulated with *Actinomyces* spp. at 0, 48, 72, and 96 h of culture. (**C**) CD4^+^ cells in controls and patients with primary infections, stimulated with *Streptococcus* spp. at 0, 48, 72, and 96 h of culture. Data correspond to the median and interquartile range, *n* = 25, * *p* < 0.05.

**Figure 4 ijms-24-16853-f004:**
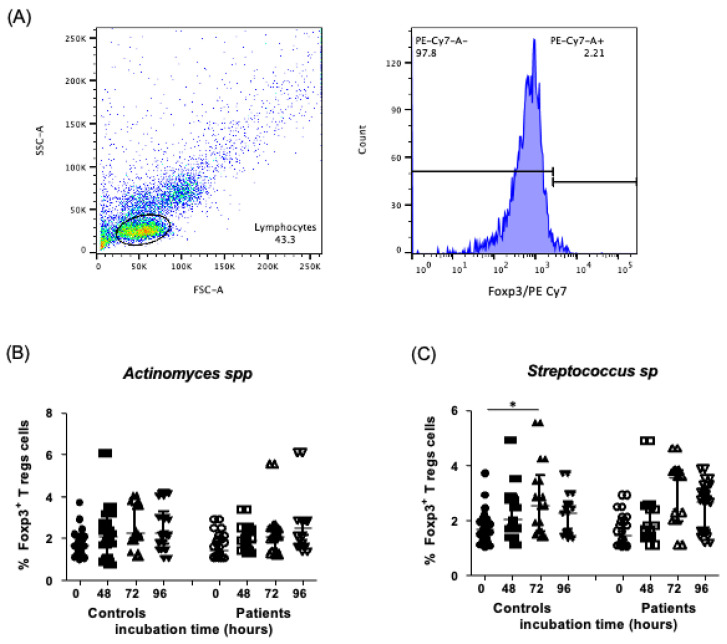
Percent of Foxp3^+^ T regulatory cells after stimulation with endodontic microorganisms. The PBMC and endodontic bacteria cultures were assessed over several durations of incubation and stained via flow cytometry analysis, as indicated in the Section 4. (**A**) Representative dot plots of a healthy subject (control) stimulated with endodontic microorganisms, where CD19^+^ lymphocytes stained with phycoerythrin Cy 7 (PE-Cy7) are shown. (**B**) Percent of Foxp3^+^ T regulatory cells in controls and patients with primary infections, stimulated with *Actinomyces* spp. at 0, 48, 72, and 96 h of incubation. (**C**) Percent of Foxp3^+^ T regulatory cells in controls and patients with primary infections, stimulated with *Streptococcus* spp. at 0, 48, 72, and 96 h of incubation. Data correspond to the median and interquartile range, *n* = 25, * *p* < 0.05.

**Figure 5 ijms-24-16853-f005:**
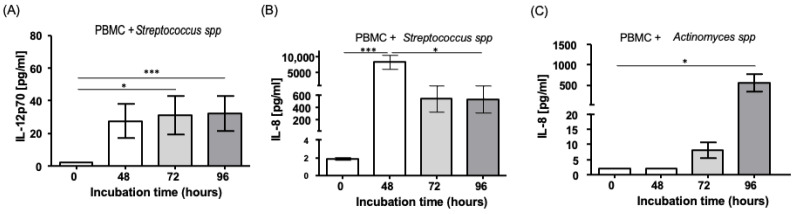
Levels of inflammatory cytokines released in PBMC from patients with primary infections, after stimulation with endodontic microorganisms. Supernatants of PBMC culture in the presence of endodontic bacteria were obtained and assayed for the presence of the indicated cytokines under flow cytometry analysis, as indicated in the Section 4. (**A**) Concentration of IL-12p70 in pg/mL under specific cell culture conditions. (**B**) Concentration of IL-8 in pg/mL under specific cell culture conditions. (**C**) Concentration of IL-8 in pg/mL under specific cell culture conditions. Data correspond to the mean and standard deviation, *n* = 10, * *p* < 0.05. *** *p* < 0.001.

## Data Availability

The data presented in this study are available within the article.

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
