# Peer review of "Increased IL-12p70 and IL-8 Produced by Monocytes in Response to Streptococcus spp. and Actinomyces spp. Causals of Endodontic Primary Infections"

_ijms, 2023, doi:10.3390/ijms242316853_

Round 1

Reviewer 1 Report

Comments and Suggestions for Authors

Thank you for the opportunity to review this article on the causative bacteria of primary endodontic infections and the cytokines produced by monocytes. It is explained in detail and is useful for summarizing knowledge. There are many unknowns in this field, but I have a few questions, and I think readers would understand more easily if authors could add explanations.

It is necessary to clearly state what should be clarified in this study. It is also necessary to clearly state where the purpose of the research is to reveal unknown areas.

Figures 1-4 are small and unclear. The letters should be enlarged to make it easier to understand.

P9L357 In the discussion section, it is discussed that Streptococcus spp. and Actinomyces spp. specifically activate monocytes, but what possible mechanism should be indicated. Is there any other pathway besides Toll-like receptors?

P10L409 These results are contrary to what we expected since the B lymphocytes are the first cells to act from an adaptive response more effective than the innate response.” The reasons for this consideration should be explained in more detail.

What statements can be generalized from this study? Should be clearly stated at the end of the discussion section.

Reviewer 2 Report

Comments and Suggestions for Authors

Dear Authors,

It was a pleasure to read your article. I believe your paper might be interesting to readers from the clinical field. Your paper is well written and organized.

However, there are some scopes to improve the quality of the manuscript. The reviewer would like to suggest the following revision in the manuscript to make it suitable for publication. 

The aim of this technical note "Increased IL-12p70 and IL-8 produced by monocytes in response to Streptococcus spp and Actinomyces spp causals of endodontic primary infections" was to evaluate the effect of endodontic-causative microorganisms of primary infections on mononuclear cells such as CD14+, CD4+ , CD8+ , CD19+ , Tregs Foxp3+.

The text should be prepared according to MDPI formatting. Use a template. 

Minor editing of English language required

Punctuation should be corrected.

Standardize text structure and alignment according to guidelines.

Material and methods.

p value should be written in italics.

Results

Figures 1, 2, 3, 4, 5 - should be larger

Discussion

Add study limitations in the discussion.

line 465 - skip to next page

Prepare references according to MDPI guidelines.

 Reconsider after major revision (control missing in some experiments)

Comments on the Quality of English Language

Minor editing of English language required

Reviewer 3 Report

Comments and Suggestions for Authors

This manuscript assessed the effects of the responsible microorganisms (mainly Streptococcus species and Actinomyces species) of endodontic primary infections on peripheral blood mononuclear cells. The cellular expression of CD14+, CD4+, CD8+, CD19+, and Tregs Foxp3+ was also analyzed in twenty patients with primary endodontic infections and twenty-five healthy subjects. The manuscript contains six keywords, five figures with different sections (Fig. 1 A to F; Fig. 2 A to E; Fig. 3 A to C; Fig. 4 A to C; Fig. 5 A to C, and thirty-eight references. Overall, it is a correct and well-conducted paper.

General comments
This study observed an increase in the percentage of CD14+ cells in endodontic patients in comparison to healthy controls. It highlights the role exerted by the innate immune system in the pathogenesis of endodontic primary infections in response to Streptococcus and/or Actinomyces species infections.
The study is methodologically well-established. The data management is appropriate according to the approach of the study. The results are well presented, being easy to read and interpret them. In the discussion section, the results of this study are adequately contrasted with those obtained by other researchers. A justifying explanation of the results is also provided. The manuscript also includes an appropriate concluding section.

Some further remarks are made on different sections of the manuscript.

Keywords
The manuscript presents six keywords. Please, avoid the use of abbreviations as keywords. For keywords, where possible, please use Medical Subject Headings terms (MeSH Terms). Strictly, only “monocytes” is a MeSH term. Some alternative MeSH terms proposed could be “interleukin 8” better than “IL-8”; “Streptococcus” rather than “Streptococcus spp”; “Actinomyces” instead of “Actinomyces spp”. Nevertheless, these suggestions about keywords are optional, not mandatory.

Other manuscript sections
Figures 1A, 1E, 1F, 2A, 3A, 4A, 5B, and 5C are not cited in the text of the manuscript. Please, include them. Moreover, reference number 29 is not cited in the text either.

References
Total number of the manuscript references: 38.
The reference format does not exactly match the journal’s reference format (ACS style guide). According to the journal’s guidelines, references should be described as follows:
1. Author 1, A.B.; Author 2, C.D. Title of the article. Abbreviated Journal Name Year, Volume, page range.

For further information about the reference format proposed by the journal, please, consult the following link: https://www.mdpi.com/journal/ijms/instructions

Figures
Total number of the manuscript figures: 5.
The figures have appropriate figure legends.

Round 2

Reviewer 1 Report

Comments and Suggestions for Authors

That paper has been properly revised.

Reviewer 2 Report

Comments and Suggestions for Authors

Limitations should be inserted after line 392.

Article can be accepted after Editor decision.
